# A Near-Ultraviolet Photodetector Based on the TaC: Cu/4 H Silicon Carbide Heterostructure

Salah Abdo, Khalil As'ham, Ambali Alade Odebowale ⓘ, Sanjida Akter ⓘ, Amer Abdulghani ⓘ, Ibrahim A. M. Al Ani ⓘ, Haroldo Hattori ⓘ and Andrey E. Miroshnichenko *ⓘ

School of Engineering and Information Technology, University of New South Wales at Canberra, Northcott Drive, Canberra, ACT 2610, Australia; s.abdo@unsw.edu.au (S.A.); k.asham@unsw.edu.au (K.A.); a.odebowale@unsw.edu.au (A.A.O.); sanjida.akter@unsw.edu.au (S.A.); a.abdulghani@unsw.edu.au (A.A.); h.hattori@unsw.edu.au (H.H.)

* Correspondence: andrey.miroshnichenko@unsw.edu.au; Tel.: +61-(2)-51145182

**Featured Application: Using the proposed photodetector (TaC: Cu/4H silicon carbide (SiC)) could broaden the use of 4H SiC-based devices beyond traditional UV detection, extending into near-ultraviolet and visible light regions, particularly at 405 nm. The high responsivity and photocurrent make it suitable for high-temperature optoelectronic applications that require reliable and efficient light detection across a wider range of wavelengths. Its possible applications include environmental monitoring, flame detection, and UV-visible optical sensing, addressing the demand for high-performance photodetection in extreme conditions.**

**Abstract:** Photodetectors (PDs) based on 4H silicon carbide (SiC) have garnered significant interest due to their exceptional optoelectronic properties. However, their photoresponse is typically restricted to the ultraviolet (UV) region, with limited light absorption beyond 380 nm, which constrains their utility in visible light detection applications. To overcome this limitation, an efficient photodetector was developed using an alloy with TaC (80%) and Cu (20%) on a 4H n-type SiC substrate, enabling effective light detection at 405 nm. The device exhibited high performance with a high photoresponsivity of 1.66 $AW^{-1}$ and a specific detectivity of $2.69 \times 10^8$ Jones at 405 nm. The superior performance of the device is ascribed to the enhanced electrical conductivity and optical absorption of the TaC: Cu layer on the 4H SiC substrate, particularly in the near-ultraviolet region. This photodetector combines ease of fabrication with significant performance improvements, expanding the potential applications of 4H SiC in high-temperature optoelectronics. It also introduces a promising pathway for enhancing 4H SiC-based photodetection capabilities across broader spectral ranges.

**Keywords:** 4H silicon carbide (SiC); tantalum carbide (TaC); near ultraviolet (NUV) photodetector; responsivity

## 1. Introduction

Photodetectors are optoelectronic devices that can precisely detect light and transform it into an electric current [1,2]. Their detection has extensively received much interest due to their wide area of applications, such as artificial vision, military, optical communications, industrial sensing, biological research, and medical imaging [3–7].

In recent years, wide bandgap materials such as gallium nitride (GaN) [8,9], zinc oxide (ZnO) [10–12], diamonds [13], aluminum gallium nitride (AlGaN) [14], gallium

oxide ($Ga_2O_3$) [15–17], and silicon carbide (SiC) [18,19] have received much attention in photodetector applications. These photodetectors can efficiently absorb ultraviolet (UV) light, and they are known for their robustness, low and dark current, high-speed response, high working temperature, and lower noise levels [20,21]. Among these wide bandgap semiconductors, SiC has garnered much interest in different areas of applications owing to its enormous electric field strength, high bond strength, and high electron saturation velocity [22–24]. Recently, it has been extensively harnessed for building ultraviolet photodetectors thanks to its advanced manufacturing technology, high thermal conductivity (3.7 W/cm·K), low intrinsic carrier concentrations, and high radiation resistance [25–28].

Many polytypes of SiC exist, including 3C (β),4H, and 6H (α) [29]. Among these SiC polytypes, 4H SiC with a wide bandgap of 3.26 eV was exploited for building ultraviolet photodetectors [30]. However, these photodetectors cannot absorb or respond to light in the visible region since the cut-off wavelength of the SiC is 380 nm [2,31]. Therefore, many materials were integrated with SiC to extend its spectral photoresponse from the ultraviolet to the near ultraviolet region and visible region. For example, Qian et al. integrated 4H-SiC with transition metal dichalcogenides (TMDCs) such as ($SnS_2$). $SnS_2$ was first fabricated on $SiO_2$ using the mechanical exfoliation method and then transferred to the SiC substrate using a dry transfer process. Even though this device showcases a high responsivity of $2.42 \times 10^4$ A/W, the time response is still relatively high at around 17 ms. In addition, the fabrication of this device using mechanical exfoliation is restricted for research and prototypes and not suitable for large-scale applications. Amed et al. fabricated a single B-doped 3C-SiC nanobelt which detects light over 380 nm [32]. The device also demonstrated excellent responsivity and was able to work well at high temperatures; however, its time response was a bit high. Recently, Haroldo et al. integrated the tantalum boride with 6 H SiC, forming a heterojunction. This device displayed a responsivity of 2.9 A/W and a time response of 260.5 ns at 504 nm [31]. Despite having excellent reported responsibility, the thickness of this device is too high at about 540 nm, which is a bit expensive to fabricate. Therefore, a new material that can boost the spectral photoresponse of SiC with a low response time through a facile fabrication process is of great importance.

Refractory transitional metal carbides such as tantalum carbide (TaC) have gained much interest as an ultra-high temperature ceramic for applications in harsh environments. It is composed of tantalum (Ta) and carbon (C) atoms produced through a reaction between tantalum metal and carbon at elevated temperatures. This material is known for its high melting point (3786 °C) [33], outstanding hardness (13.5–20 GPa) [34], remarkable resistance to chemical attacks [35], high density (14.6 g/cm$^3$) [36], specific heat ($5.44 \times 10^5$ J/kg) [37], elastic modulus (~477 GPa) [38], bending strength of 340–400 MPa [39], and small thermal expansion coefficient of $6 \times 10^6$ (1/K) [40]. These remarkable characteristics and chemical properties are ascribed to the mixed covalent metallic bond [41,42], making it an appealing material for different applications, such as cutting tools, aerospace components, and rocket propulsion [43]. In addition to its excellent mechanical properties, tantalum carbide has a low work function of ~3.14 eV [44] and low resistivity (42.1 μΩ cm at 25 °C) [45]. This property makes it an attractive material for applications that demand high working temperatures, such as metal contact for optoelectronic devices [46,47] and heating elements [48]. Moreover, tantalum carbide exhibits an inherent wavelength selectivity, making it an attractive material for applications requiring high temperatures, such as solar absorbers [49].

On the other hand, copper metal with a purity of 99.95% possesses outstanding properties such as a relatively high melting point of (1085 °C), high mechanical strength, excellent electrical conductivity, remarkable thermal conductivity, and high oxidation resistivity [50–52]. It also has sufficient hardness when functioning at temperatures below 100 °C [53,54]. Therefore, owing to all these properties together with its easy fabrication process [55], copper is a suitable material to form a composite with other materials and is applicable in different photodetectors [56,57].

Inspired by the properties of SiC, TaC, and copper materials, the integration of these materials in photodetector design can enhance the performance of 4H-SiC. Consequently, a near-ultraviolet photodetector based on a TaC (80%)/Cu (20%)/4H-SiC heterostructure was proposed and fabricated using the co-sputtering method, achieving a notable responsivity of 1.66 A/W and $2.69 \times 10^8$ Jones at a wavelength of 405 nm. The device also extends its photoresponse to the visible region, demonstrating its potential for applications in astronomical observations, DNA analysis, and fluorescence microscopy.

## 2. Materials and Methods

### 2.1. Material Preparation

This section describes the fabrication process of the TaC: Cu/4H SiC photodetector. The n-type 4H SiC substrate, with a doping level of $1 \times 10^{18}$ cm$^{-3}$, a thickness of 350 μm, and dimensions of 10 mm × 10 mm, was procured from MSE Supplies (Tucson, AZ, USA). High-purity tantalum carbide (TaC) with a 2:1 atomic ratio and copper (Cu) targets, each with 99.9% purity and a 4-inch diameter, were acquired from Loyal Target Tech. Prior to deposition, the 4H SiC sample underwent rigorous cleaning to eliminate surface contamination and native oxides. This involved the immersion of the 4H SiC substrate in an ultrasonic bath with acetone and deionized water for twenty minutes, followed by treatment with buffered oxide etchant. The cleaned substrate was then dried using high-purity nitrogen gas.

### 2.2. Fabrication of the Device

The deposition of the TaC (80%): Cu (20%) alloy was conducted using a commercial co-sputtering system (AJA ATC 2200, Scituate, MA, USA) equipped with six sputtering sources. The TaC target was powered by a 300 W radio frequency (RF) supply, while the Cu target was driven by a direct current (DC) power supply. To prevent surface contamination, both targets were pre-sputtered for two minutes without the substrate in the chamber. Afterward, the cleaned SiC substrate was inserted, pre-heated at 200 °C for 45 min to remove residual air and moisture and then allowed to cool to room temperature (approximately 30 °C). During sputtering, the chamber pressure was held at $1 \times 10^{-4}$ mbar, and 99.99% pure argon gas was introduced at a flow rate of 20 sccm. The target-to-substrate distance was set to 9 cm to ensure a uniform deposition. TaC and Cu were co-deposited onto the SiC substrate over 37 min, forming a TaC: Cu/4H-SiC heterostructure. The copper doping concentration was adjusted by regulating the DC power supply to the Cu target. The deposition rate of the TaC: Cu alloy was measured using a surface profiler, yielding a rate of 4.5 nm/min.

Subsequently, the photodetector was fabricated by depositing two aluminum (Al) electrodes, each 150 nm thick. One electrode was applied to the top surface of the alloy film, while the other was placed on the bottom side of the substrate. This deposition was carried out using a Temescal BJD-2000 (FerroTech, Livermore, CA, USA) electron beam evaporator under ambient conditions at a fixed temperature of 30 °C. The overall fabrication process of the proposed device is illustrated in Figure 1.

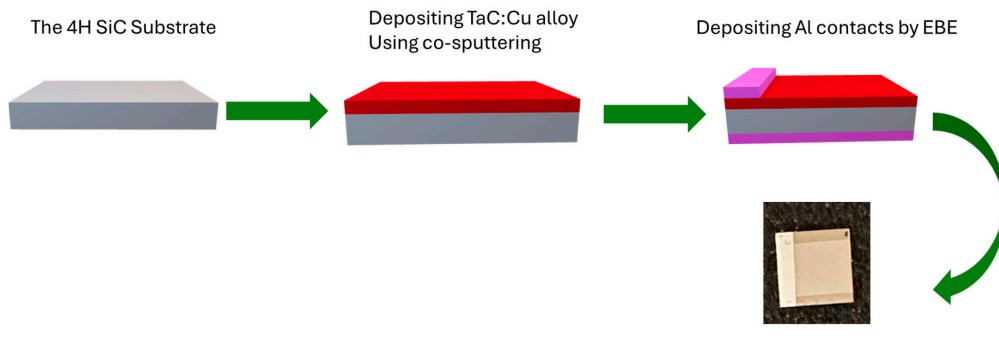

The 4H SiC Substrate

Depositing TaC:Cu alloy Using co-sputtering

Depositing Al contacts by EBE

Photography of the final device

**Figure 1.** The fabrication process of the proposed photodetector TaC: Cu/4 H SiC.

*2.3. Characterization*

The deposition rate and film thickness of the TaC: Cu alloy on the glass substrate were measured using a Dektak (Bruker Corporation, Hillsboro, OR, USA) surface profiler, ensuring a precise assessment of material growth. Surface morphology analysis was conducted with a scanning electron microscope (SEM, FEI Verios) at a working distance of 5 mm, with operating voltages ranging from 5 kV to 15 kV. The elemental composition was evaluated through energy-dispersive X-ray spectroscopy (EDX), utilizing an Oxford INCA X-act detector integrated within the SEM system. To investigate the cross-sectional architecture of the fabricated photodetector (TaC: Cu/4H-SiC), a focused ion beam (FIB) system (FEI Helios 600 NanoLab (FEI, Hillsboro, OR, USA)) was employed. This advanced tool, featuring high-resolution SEM imaging (0.9 nm) and ion beam capabilities, enabled detailed observation of the film's cross-sectional morphology.

The dielectric properties and refractive index of the TaC: Cu alloy were determined using a JA Woollam M2000D (J.A. Woollam, Lincoln, NE, USA) ellipsometer, which operates over a broad spectral range spanning from 190 nm to 1690 nm. Optical characteristics, such as absorption and transmission, were simulated with the Lumerical FDTD software (2024R2.1) packages. These simulations leveraged the thickness and refractive index data obtained through ellipsometry. The electrical properties of the TaC: Cu alloy deposited on a glass substrate were assessed using a four-point probe system from Ossila.

The photoresponse properties of the proposed device were thoroughly assessed using state-of-the-art instrumentation. An optical light source from Thorlabs S1FC405 (Thorlabs, Newton, NJ, USA) laser, emitting light at a wavelength of 405 nm, was used as the light input for the device. Electrical measurements, including current–voltage (I-V) profiling, were carried out using a Keithley 2450 source meter. To evaluate the time response, a Q-switched laser with a pulse duration of 10 ns was employed, ensuring precise analysis of the device's dynamic behavior.

## 3. Results and Discussions

*3.1. Morphological Properties*

The images of surface morphology, elemental composition and cross-sections of the TaC: Cu alloy deposited on a glass and 4H SiC substrates are illustrated in Figures 2 and 3. Figure 2a presents an SEM image that reveals the surface characteristics of the alloy. The image predominantly displays a relatively grainy surface. Figure 2b provides a cross-sectional view of the TaC: Cu alloy deposited on the glass substrate, where the measured thickness is 126.5 nm.

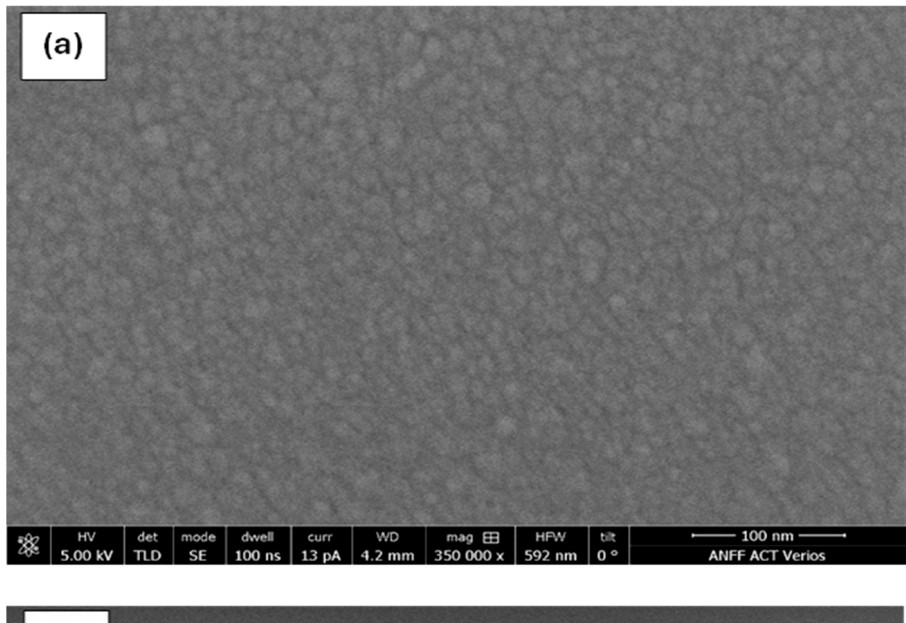

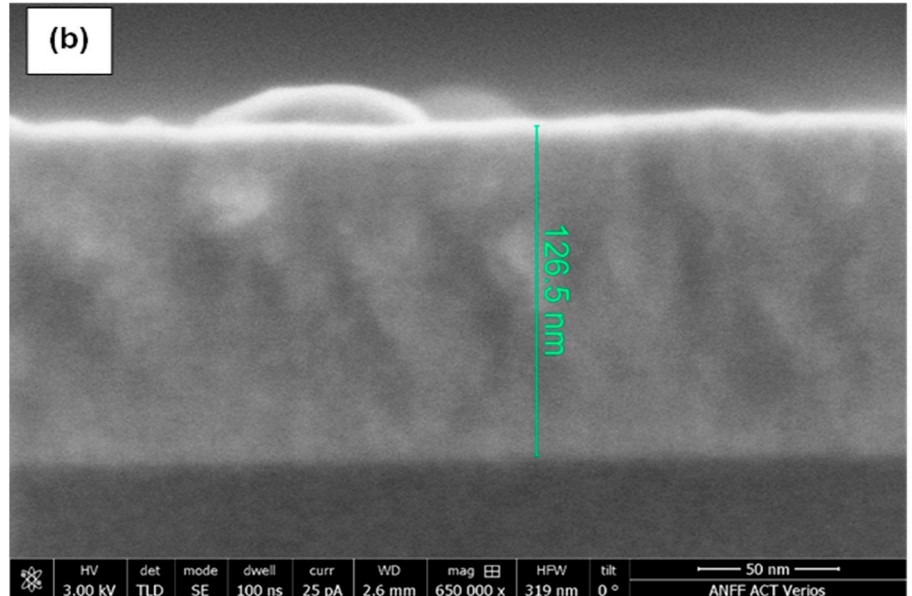

**Figure 2.** (**a**) The top-view SEM image for the TaC: Cu alloy (**b**) The alloy cross-section on a glass substrate.

The elemental composition of the deposited TaC: Cu alloy was analyzed using energy-dispersive X-ray spectroscopy (EDX) to quantify the relative contributions of each element. Figure 3a shows the EDX spectra of the TaC: Cu alloy, clearly identifying the elemental composition. A prominent peak at approximately 0.89 keV corresponds to tantalum (Ta), confirming its role as the primary component in the TaC film, followed by copper (Cu) and oxygen (O). Carbon (C) represents the element with the lowest content in the alloy. The presence of Cu, carbon, and tantalum indicates the formation of the tantalum carbide (TaC: Cu) alloy, with potential surface oxidation, which is common in thin-film deposition processes.

After characterizing the surface morphology and elemental composition of the TaC: Cu alloy, the final device was deposited onto a 4H-SiC substrate via sputtering for 37 min. Figure 3b presents a schematic of the fabricated device, where the upper and lower electrodes are depicted in dark contrast. To investigate the cross-section of the photodetector, a focused ion beam (FIB) system integrated with scanning electron microscopy (SEM) was employed. The FIB system facilitated precise localized etching, enabling the acquisition of high-resolution cross-sectional images. During the scanning process, the sample was

meticulously positioned 2.5 nm from the FIB lens, and a voltage of 30 kV was applied to ensure optimal imaging. Figure 3c showcases the cross-sectional image of the fabricated device on the 4H n-type SiC substrate, clearly delineating three distinct layers: the protective platinum (Pt) top layer, designed to shield the sample from FIB-induced damage, the TaC: Cu layer with a measured thickness of 168 nm, and the 4H n-type SiC substrate with black color.

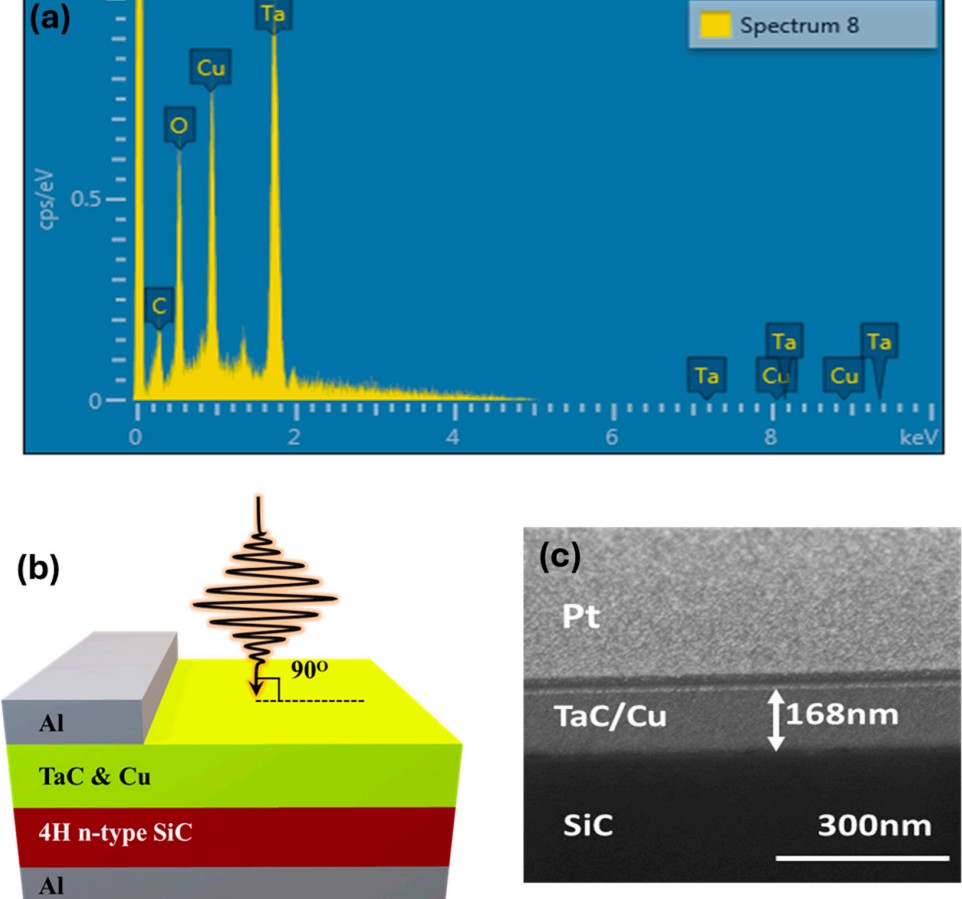

**Figure 3.** (**a**) EDX elemental composition of TaC: Cu on a glass substrate (**b**) The schematic structure of the fabricated device (**c**) FIB image of the cross-section of the fabricated device with 300 nm scale par.

### 3.2. Electrical and Optical Properties

The resistivity and electrical conductivity measurements of the TaC: Cu alloy were obtained based on the sample deposited on a glass substrate using a four-point probe system (Ossila T2001A3 (Ossila, Sheffield, UK)). To ensure accuracy, measurements were conducted at three different positions of the sample, and the average values were calculated. The TaC: Cu alloy exhibited a resistivity of 14.1 μΩ·cm and a high electrical conductivity of 9.3 KS/m at room temperature. This notable conductivity is attributed to the enhanced charge carrier density introduced by copper doping.

The dielectric function ($\psi$) of the TaC: Cu film was characterized using variable-angle spectroscopic ellipsometry (VASE) over a spectral range from 200 nm to 1800 nm. Measurements were conducted at three distinct incident angles (55°, 65°, and 75°), and the resulting data were analyzed using a general oscillator model. The excellent agreement between the experimental data and the model-generated results, as illustrated in Figure 4a, validates the accuracy and reliability of the measurements. Furthermore, spectroscopic ellipsometry (SE) was performed over the same wavelength range, with a step size of 2 nm,

to determine the optical constants of the TaC: Cu alloy. The analysis assumed a single uniform layer and ignored the surface roughness.

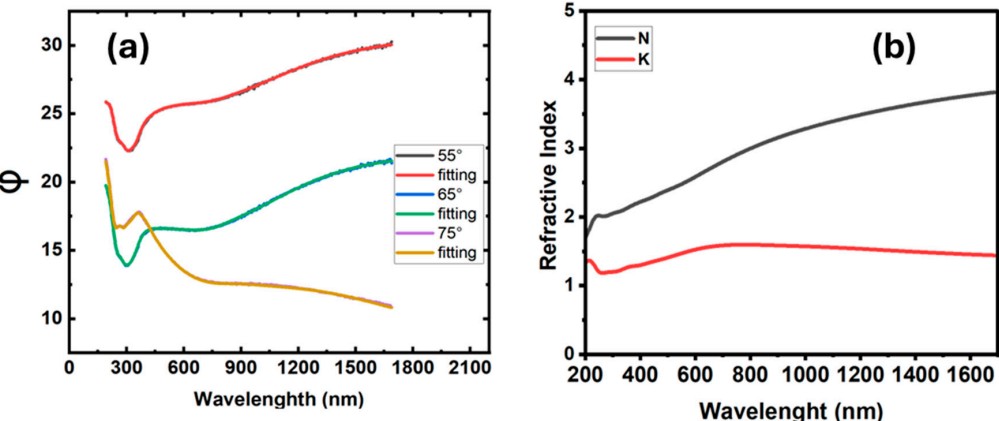

**Figure 4.** (**a**) The dielectric function ($\psi$) of the TaC: Cu alloy on a glass substrate using variable-angle spectroscopic ellipsometry (VASE) over a spectral range from 200 nm to 1800 nm. (**b**) The refractive index of the TaC: Cu alloy on a glass substrate.

The refractive index and film thickness were subsequently extracted from the ellipsometry fitting. Figure 4b depicts the real (n) and imaginary (k) components of the refractive index. The real component attains a maximum value of 3.9 at a wavelength of 1700 nm, whereas the imaginary component peaks at 1.58 at 700 nm and exhibits a gradual decline with increasing wavelength. This behavior indicates that the TaC: Cu alloy demonstrates a broad spectral detection capability, spanning from the ultraviolet (UV) to the near-infrared (NIR) regions. The film thickness, determined via ellipsometry without accounting for surface roughness, was measured to be 127 nm.

The absorption and transmission characteristics of the proposed device were analyzed through simulations utilizing the Finite-Difference Time-Domain (FDTD) method, carried out using commercial Lumerical software (2024R2.1). Figure 5a illustrates the simulated structure, while the simulation setup is depicted in Figure 5b. A plane wave source spanning the spectral range of 200–1800 nm was injected along the Z-axis, with perfectly matched layer (PML) boundary conditions applied in the Z-direction and periodic boundary conditions in the X and Y directions. Monitors recorded the transmission (T) and reflection (R) data, while absorption was calculated using the relation $1 - R - T$ [58]. The refractive index of the TaC: Cu alloy and SiC substrate were imported into the simulation based on experimental measurements and values obtained from the published literature [59]. In addition, the thickness of the TaC: Cu alloy was set to 127 nm, as determined by ellipsometry measurements. As shown in Figure 5c, the simulation revealed a pronounced absorption peak of approximately 77% at 259 nm, with absorption decreasing to 73% at 405 nm and gradually diminishing across the near-infrared (NIR) region. Despite the decline, the film exhibited measurable absorption extending beyond 1800 nm.

The simulated transmittance behavior of the TaC: Cu alloy on SiC is shown in Figure 5d. This hybrid structure exhibited near-zero transmittance in the 200–600 nm range due to high absorption in this region. However, transmittance steadily increased as the wavelength extended into the NIR regions. These findings highlight the exceptional performance of the proposed hybrid structure, emphasizing its potential to enhance operational efficiency in the ultraviolet (UV) and near-ultraviolet bands.

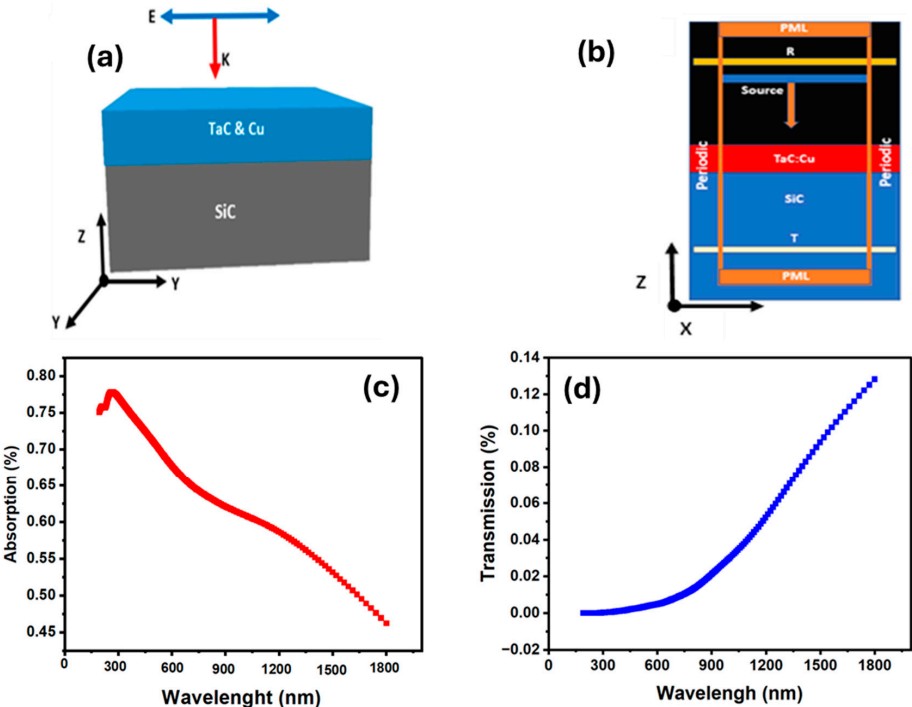

**Figure 5.** (**a**) The schematic structure of the TaC: C/4H SiC hybrid structure. (**b**) The FDTD simulation region used to simulate the optical absorption and transmittance of the TaC: Cu alloy on SiC. (**c**,**d**) The simulated optical absorption and transmission as a function of the wavelength for the TaC: Cu alloy on SiC, respectively.

### 3.3. Photoreponse Characteristics

The performance of the photoresponse device was evaluated through its current–voltage (I-V) characteristics. Figure 6 illustrates the variation in dark current with applied voltage. In the reverse bias region, the dark current increases exponentially as the voltage rises from −1.75 V to 0 V. When the bias voltage shifts to the positive range (forward bias), the current shows a rapid exponential growth from 0 V to 1.75 V.

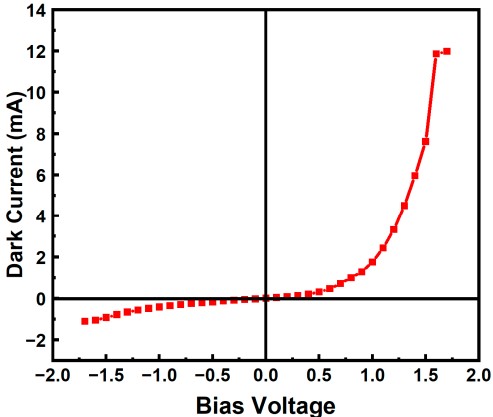

**Figure 6.** The dark current of the TaC: Cu/4H SiC heterostructure.

Figure 7a shows the relationship between the total current and applied bias voltage at different light intensities. It reveals that the total photocurrent increases linearly with light intensity under reverse bias. Furthermore, the relationship between photocurrent and light power at a reverse voltage is depicted in Figure 7b. The photocurrent initially grows linearly with increasing light power but reaches a saturation point beyond 400 μW, after which no significant increase is observed.

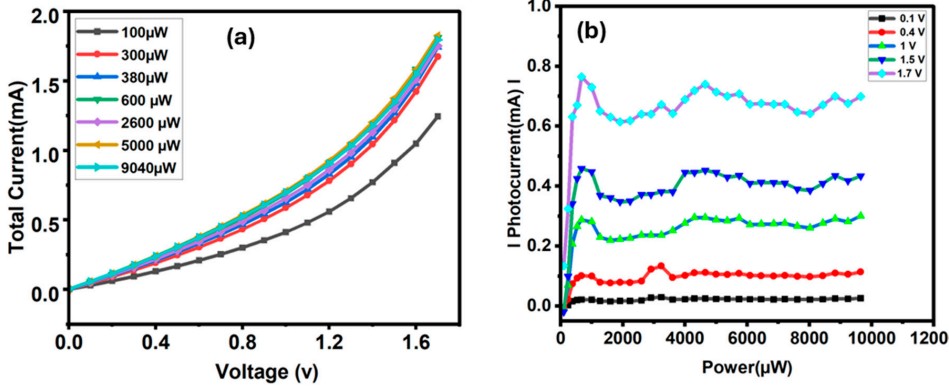

**Figure 7.** (**a**) The total illuminated current and (**b**) the photocurrent variation with power.

The responsivity of the fabricated device was calculated using the following equation [60]:

$$R = \frac{|I_{total} - I_{dark}|}{P_{optical}} \tag{1}$$

where $I_{light}$ denotes the illuminated current, $I_{dark}$ represents the dark current, and $P_{optical}$ stands for the incident optical power. Figure 8a illustrates the relationship between responsivity and incident optical power. The responsivity exhibits significant variation with changes in incident power. Initially, it increases with rising incident power and stabilizes at approximately 380 µW. Beyond this threshold, the responsivity decreases sharply as the incident power continues to rise. This behavior is attributed to the saturation of the photocurrent at higher power levels, which reduces the responsivity. The measured responsivity values at applied voltages of 0.4 V, 1 V, and 1.75 V were 0.195, 0.544, and 1.66 A/W, respectively. Figure 8b presents the normalized responsivity of the photodetector across different wavelengths. The normalized responsivity was measured using four LED sources and relatively normalized to the responsivity at 405 nm. The graph includes three curves corresponding to different applied voltages: the black line for 0.4 V, the red line for 1 V, and the blue line for 1.75 V. The responsivity reaches its peak value of 1 at 405 nm and gradually decreases to 0.915 at 650 nm.

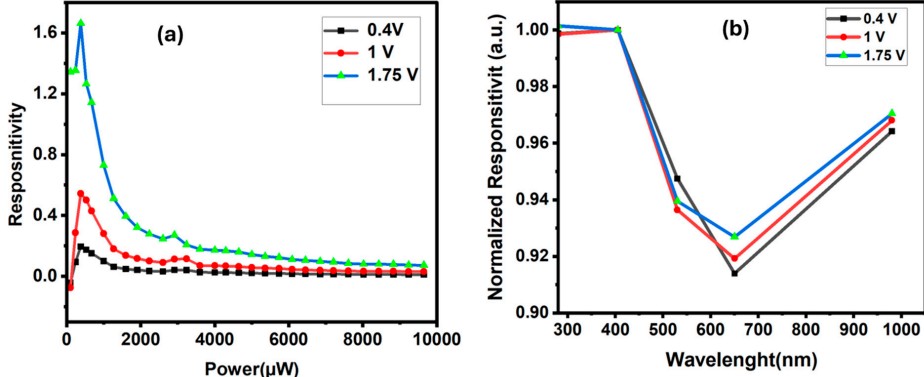

**Figure 8.** (**a**) Photoresponsivity as a function of input power. (**b**) The normalized responsivity plotted against wavelength.

Another critical parameter for evaluating photodetector performance is detectivity ($D^*$), which was determined using the following equation [61]:

$$D^* = R\sqrt{\frac{A_e}{2qI_{dark}}} \tag{2}$$

where $R$ refers to the responsivity, $A_e$ represents the area under illumination, $q$ is the electron charge, and $I_{dark}$ denotes the dark current. By applying these parameters to the above equation, the highest obtained detectivity was $2.69 \times 10^8$ Jones at 1.75 V, and this dropped with the increase in the incident power.

To evaluate the time response of the device, a Q-switched laser source with a peak power of 1500 W, a pulse duration of 10 ns, and a repetition rate of 1 kHz was employed. An optical attenuator was utilized to decrease the average beam power to 1 mW. Under these conditions, with a beam diameter of 2 mm, the resulting response spectrum is depicted in Figure 9. The envelope of the pulse oscillation spans approximately 260.7 μs. The response time was determined using the following formula [31]:

$$\tau_{total} = \sqrt{\tau_{initial}^2 + \tau_{response}^2} \tag{3}$$

Consequently, the cut-off frequency can be calculated as follows:

$$f_{cutoff} = \frac{1}{2\pi\tau_{response}} \tag{4}$$

where $\tau_{respose}$ is the response time with a value of 260.2 μs, and $\tau_{total}$ is the total response time with a value of 260.5 μs. The $\tau_{initial}$ is the pulse duration of the exciting laser source, which is equal to 10 ns, and the cut-off frequency is 611 kHz.

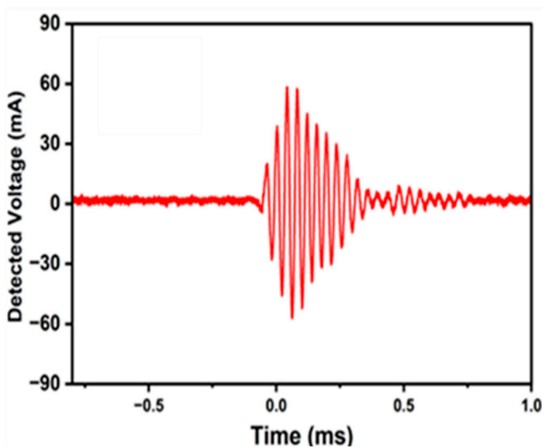

**Figure 9.** Time response of the fabricated device upon excitation with a 10 ns Q-switched pulse.

A comparative analysis of the proposed photodetector with previously reported devices is summarized in Table 1. ZnO-based photodetectors, as discussed in [11,62,63], demonstrate low and dark currents but suffer from significantly reduced responsivity and slower response times. Similarly, the $Ga_2O_3$/GaN photodetector presented in Ref. [64] achieves a responsivity of $5 \times 10^2$ mAW$^{-1}$ at 2 V, which can be markedly improved through field-enhanced exciton ionization at higher voltages. For instance, at 16 V, the responsivity increases dramatically to approximately $2.6 \times 10^6$ mAW$^{-1}$. However, this enhancement comes at the cost of a very narrow response bandwidth of around 4 nm and a sluggish response time of 0.1 s due to the ionization process. Additionally, a boron-doped 3C-SiC nanobelt photodetector reported in Ref. [19] recorded a remarkable responsivity of $6.37 \times 10^8$ mA/W and a response time of 50 ms at a 405 nm operating wavelength. In comparison, even though the SiC/TaB-based photodetector described in Ref. [31] achieved a responsivity of $2.9 \times 10^3$ mA/W with a faster response time of 260.5 ns, a thick layer of TaB is needed to achieve such a performance, which contributes to the higher cost of photodetector design. In contrast, the proposed TaC: Cu/4H n-SiC photodetector outperforms many of these alternatives by delivering a balanced performance. It achieves

a responsivity of 1.66 AW$^{-1}$, maintains a low and dark current of 1.1 mA, and demonstrates a rapid response time of 260.2 µs, making it a compelling candidate for practical photodetection applications.

**Table 1.** Comparison of different photodetectors.

| Structure | Wavelength (nm) | Responsivity (mA/W) | Dark Current (mA) | Response Time (s) | Reference |
|---|---|---|---|---|---|
| 6H SiC/TaB | 405 | $2.9 \times 10^3$ | 500 at −1 V | $260.5 \times 10^{-9}$ | [31] |
| PANI ZnO film | 380 at −3 V | 37.0 | $0.8 \times 10^{-10}$ | $50 \times 10^{-6}$ | [62] |
| Ga$_2$O$_3$/GaN | 360 | $5 \times 10^2$ at 2 V $2.6 \times 10^6$ at 16 V | - | 0.1 | [64] |
| Pt/Ni at nanowire/ZnO film | 312 | 17.0 | - | 2.9 | [11] |
| Au/ZnO | 325 | 0.485 | $2 \times 10^{-3}$ | $12 \times 10^{-3}$ | [63] |
| B-doped 3C-SiC nanobelts | 405 | $6.37 \times 10^8$ at 5 V | $4.37 \times 10^{-4}$ | $50 \times 10^{-3}$ | [19] |
| Ag-4H-SiC nanohole-Ag | 375 | 824 | - | 1.022 | [65] |
| 2D graphene/1D 4H-SiC NWAs | 365 | 9.27 at 0 V $9.27 \times 10^3$ at 5 V | $4 \times 10^{-9}$ | $24.89 \times 10^{-3}$ | [66] |
| 4H-SiC/SiO$_2$/Si (100) | 280 | 29 | $1.6 \times 10^3$ | - | [67] |
| TaC: Cu/4H n-type SiC | 405 | $1.66 \times 10^3$ | 1.1 at −1.75 V | $260.2 \times 10^{-6}$ | This work |

## 4. Conclusions

This study presents a TaC: Cu/4H SiC heterostructure photodetector with remarkable performance for near-ultraviolet (NUV) light detection, extending its application into the visible spectrum. The device achieved high responsivity (1.66 A/W at 405 nm) and a rapid response time, demonstrating its suitability for high-temperature and extreme-environment applications. By leveraging the unique optoelectronic properties of TaC and Cu, combined with 4H SiC, this photodetector addresses the limitations of conventional SiC-based photodetectors. The results establish the potential for its scalable fabrication and practical deployment in fields such as environmental monitoring, flame detection, and optical sensing, marking a significant step forward in high-performance photodetection technology.

**Author Contributions:** Conceptualization, S.A. (Salah Abdo) and H.H.; Methodology, S.A. (Salah Abdo) and H.H.; Software, S.A. (Salah Abdo); Validation, H.H.; Formal analysis, S.A. (Salah Abdo); Investigation, S.A. (Salah Abdo) and S.A. (Sanjida Akter); Resources, S.A. (Salah Abdo); Writing—original draft, S.A. (Salah Abdo); Writing—review & editing, K.A., A.A.O., A.A. and I.A.M.A.A.; Supervision, H.H. and A.E.M.; Project administration, H.H.; Funding acquisition, H.H. and A.E.M. All authors have read and agreed to the published version of the manuscript.

**Funding:** This research was funded through the Australian Research Council Discovery Project grant (DP200101353).

**Institutional Review Board Statement:** Not applicable.

**Informed Consent Statement:** Not applicable.

**Data Availability Statement:** The data that support the findings of this study are available from the corresponding author upon reasonable request.

**Acknowledgments:** The authors express their gratitude to the Australian National Fabrication Facilities (ANFF), ACT node.

**Conflicts of Interest:** The authors declare no conflicts of interest.

## Abbreviations

The following abbreviations are used in this manuscript:

| | |
|---|---|
| PDs | photodetectors (PDs) |
| SiC | silicon carbide |
| TaC | tantalum carbide |
| Cu | Copper |
| GaAs | gallium arsenide |
| UV | ultraviolet |
| GaN | gallium nitride |
| ZnO | zinc oxide |
| AlGaN | aluminum gallium nitride |
| $Ga_2O_3$ | gallium oxide |
| TMC | transition metal carbides |
| RF | radio frequency |
| DC | direct current |
| SEM | scanning electron microscope |
| FIB | focused ion beam |
| EDX | energy-dispersive X-ray spectroscopy |
| SE | spectroscopic ellipsometry |
| NUV | near-ultraviolet |
| FDTD | Finite-Difference Time-Domain |

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
