# Peer review of "A Near-Ultraviolet Photodetector Based on the TaC: Cu/4 H Silicon Carbide Heterostructure"

_applsci, doi:10.3390/app15020970_

Round 1

Reviewer 1 Report

Comments and Suggestions for Authors

This paper presents the development and characterization of a near-ultraviolet photodetector based on a TaC:Cu/4H silicon carbide (SiC) heterostructure. The device is designed to overcome the limitations of traditional 4H-SiC photodetectors, which are typically restricted to the ultraviolet (UV) region and have limited light absorption beyond 380 nm. This paper is well written and the data and discussion are convincing. I recommend publication of this paper after some minor revision.

1)    The literature review could be strengthened by including more recent studies on the integration of TaC with other semiconductor materials for photodetection applications.

2)    The photoresponse characteristics, such as responsivity and detectivity, should be compared with state-of-the-art devices in the literature to highlight the improvements achieved.

3)    The time response measurements could be further validated by performing additional tests at different wavelengths and power levels.

Reviewer 2 Report

Comments and Suggestions for Authors

It was a very interesting paper on using alloyed TaC:Cu/4H-SiC heterostructures for near-UV photodetection. Here are some suggestions to improve the quality of the paper: 

1. It will be interesting to see how the detectivity and responsivity of the fabricated photodetector change with operating temperature. Although this was mentioned as a motivation for such a device, it was not explored experimentally.

2. The paper does not go into how the copper physically modulates the electronic properties of TaC. Some electronic band-structure of the alloy and the junction of the photodiode will add more scientific value.  
